# Multivariable Signal Processing for Characterization of Failure Modes in Thin-Ply Hybrid Laminates Using Acoustic Emission Sensors

**DOI:** 10.3390/s23115244

**Published:** 2023-05-31

**Authors:** Sakineh Fotouhi, Maher Assaad, Mohamed Nasor, Ahmed Imran, Akram Ashames, Mohammad Fotouhi

**Affiliations:** 1School of Engineering, University of Glasgow, Glasgow G12 8QQ, UK; sakineh.fotouhi@glasgow.ac.uk; 2Department of Electrical and Computer Engineering, College of Engineering and IT, Ajman University, Ajman P.O. Box 346, United Arab Emirates; 3Department of Biomedical Engineering, College of Engineering and IT, Ajman University, Ajman P.O. Box 346, United Arab Emirates; 4College of Pharmacy and Health Sciences, Ajman University, Ajman P.O. Box 346, United Arab Emirates; 5Medical and Bio-Allied Health Sciences Research Centre, Ajman University, Ajman P.O. Box 346, United Arab Emirates; 6Faculty of Civil Engineering and Geosciences, Delft University of Technology, 2628 CD Delft, The Netherlands

**Keywords:** multivariable analysis, acoustic emission, fragmentation, carbon/glass hybrids

## Abstract

The aim of this study was to find the correlation between failure modes and acoustic emission (AE) events in a comprehensive range of thin-ply pseudo-ductile hybrid composite laminates when loaded under uniaxial tension. The investigated hybrid laminates were Unidirectional (UD), Quasi-Isotropic (QI) and open-hole QI configurations composed of S-glass and several thin carbon prepregs. The laminates exhibited stress-strain responses that follow the elastic-yielding-hardening pattern commonly observed in ductile metals. The laminates experienced different sizes of gradual failure modes of carbon ply fragmentation and dispersed delamination. To analyze the correlation between these failure modes and AE signals, a multivariable clustering method was employed using Gaussian mixture model. The clustering results and visual observations were used to determine two AE clusters, corresponding to fragmentation and delamination modes, with high amplitude, energy, and duration signals linked to fragmentation. In contrast to the common belief, there was no correlation between the high frequency signals and the carbon fibre fragmentation. The multivariable AE analysis was able to identify fibre fracture and delamination and their sequence. However, the quantitative assessment of these failure modes was influenced by the nature of failure that depends on various factors, such as stacking sequence, material properties, energy release rate, and geometry.

## 1. Introduction

Aircrafts, spacecrafts, motor sports, high-performance sports equipment, and biomaterials, such as orthopedic implants and prosthetics, are advanced applications with extensive use of high-performance fibre-reinforced plastic (FRP) composites. This is primarily because of their exceptional strength and stiffness, along with their low density. However, their relatively high material and manufacturing costs, as well as their tendency to fail abruptly and catastrophically without sufficient warning, restrict their use in applications where the risk of unexpected failure and subsequent loss of residual integrity is unacceptable.

A new type of FRP composite, called the Pseudo-Ductile Hybrid (PDH) composite, has been developed to tackle the issue of sudden and brittle failure [1,2]. PDH composites, if designed properly, can surpass traditional glass or carbon FRPs [3,4,5]. To achieve an optimal PDH design, it is necessary to carefully choose the constitutive high strain material (HSM) and low strain material (LSM) properties and their thicknesses, as suggested in [4]. Figure 1 shows the modes of failures in conventional hybrid composites, leading to a catastrophic failure: (a) a single crack that spans the entire laminate’s thickness as a result of an imbalanced thickness proportion of LSM to HSM and (b) failure of the LSM immediately followed by a single delamination. In PDH composites, achieving a gradual failure mode is preferable, where delamination is prevented and multiple fractures occur in the LSM, leading to dispersed delamination [4,5,6].

UD and quasi-isotropic (QI) configurations with different combinations of HSMs and LSMs have been proposed as PDH composites to generate the desired pseudo-ductile behaviour [4,5]. These PDHs have proven beneficial in mitigating the notch sensitivity issue that has long been a major limiting factor in conventional composite laminates [6].

It is essential to identify the failure modes that result in pseudo-ductility in order to optimize component use and design more generalized layups with gradual failure. However, characterizing these failure mechanisms can be quite difficult, especially when the failure modes are not readily discernible. This can occur in laminates that are thick, or when multiple failure modes occur simultaneously. Acoustic Emission (AE) monitoring of PDH composites can also be used to monitor the structural integrity of PDH composites for different applications, for example, orthopaedic implants and prosthetics. AE can evaluate different failure modes by recording the released strain energy in the form of mechanical waves originating from these failure modes under loading. The recorded AE signal can be defined with different features, such as frequency, amplitude, count, rise time, duration, and energy. The unique advantage of AE, as opposed to other non-destructive methods, is its ability to provide continuous structural monitoring by capturing the AE signals originating from failure initiation and propagation within the material. In this investigation, the effectiveness of the AE technique in characterizing the failure evolution and identifying the failure mode in PDH laminates is explored.

Numerous works have attempted to evaluate the correlation between AE signal parameters and failure modes in FRPs. The range of AE features, such as energy and frequency, generated from different failure modes, have been shown to differ from each other [7,8,9,10,11,12,13]. High levels of AE frequency and energy ranges are commonly linked to the failure of fibres, while moderate ranges of AE signals are often associated with delamination or debonding, and low ranges of AE signals are indicative of transverse or longitudinal matrix cracking. Some studies have found that frequency is a good AE feature to distinguish between failure modes [8,9,10], while others have used energy or amplitude of AE signals [11,12]. Previous researchers have utilized cumulative energy/count graphs to establish limits for failure progress [13,14]. However, prior research has indicated that relying on one feature of an AE signal is usually inadequate for distinguishing between various failure mechanisms. Therefore, multivariable analysis techniques, such as genetic algorithms and fuzzy c-means clustering, were employed to effectively categorize AE signals related to various modes of failure [15,16]. 

Previous studies have demonstrated the effectiveness of the AE technique in detecting failure mechanisms in UD PDH laminates when subjected to tensile and repeated quasi-static tensile loadings [17,18], using single parameter amplitude and energy distributions. However, these failure mechanisms vary in size and intensity by changing the stacking sequence and introducing discontinuity. The current study aims to assess the ability of the AE technique in identifying the initiation and accumulation of failure in a comprehensive range of thin-ply PDH composite materials under tensile loading. To achieve this aim, UD, QI, and open-hole QI PDH laminates were exposed to tensile loading, creating different sizes of LSM fragmentation and dispersed delamination, and the resulting modes of failure were monitored. Multivariable analysis tools are employed to categorize the AE signals and assess the failure mechanisms and their progression.

## 2. Experimental Procedures

### 2.1. Materials and Design

Table 1 provides the material constitutions of the UD, QI, and open-hole QI PDHs. Figure 2 illustrates the stacking sequence and the lay-up arrangements. The HSM was a UD S-glass/913epoxy prepreg provided by Hexcel (Stamford, CT, USA). Meanwhile, the LSMs were thin carbon prepregs provided by SK Chemicals (Seongnam-si, Republic of Korea), known as SkyFlex USN020A (TR30/epoxy and T300/epoxy).

To create the laminates, failure mode maps [4] were utilized to determine appropriate thicknesses of the LSM and HSM that would result in pseudo-ductile failure modes, i.e., fragmentation in the LSM and dispersed delamination [1]. The resulting failure mode maps are shown in Figure 3, which illustrate the boundaries between different regions. The phrase “carbon relative thickness” pertains to the carbon layers’ thickness in relation to the thickness of the entire laminate. The hues on the maps indicate the anticipated magnitude of pseudo-ductile strain. Based on the failure mode maps, the expected failure scenario in the investigated laminates is the fragmentation of the LSM (carbon), followed by dispersed delamination and failure of HSM (glass). It should be noted that these maps were obtained for UD and QI specimens, and failure mechanisms in open-hole QI may differ slightly.

### 2.2. Manufacturing Procedures

The specimens underwent curing at 120 °C, following the recommended temperature provided by the supplier. End tabs were woven glass/epoxy plates, with 2 mm thickness, and were attached to the specimens by a two-part Araldite 2000 A/B epoxy adhesive, with a weight fraction ratio of 100:50, provided by Huntsman. The specimens were then cured at 80 °C for 2 h in an oven. Cross-sectional micrographs confirmed that the hybrid laminates maintained good integrity, without any observed phase separation, indicating that the resin systems were chemo-mechanically compatible. Composite specimens were cut to the desired test size using a CNC milling machine. Figure 4 displays information regarding the nominal geometrical parameters and types of the investigated specimens.

### 2.3. Test Procedures

Six specimens of each configuration were tested in tension under displacement control conditions, by an Instron 8801 hydraulic test machine with a 25 kN load cell, utilizing wedge-type hydraulic grips. For the UD, QI, and open-hole QI laminates, crosshead speeds of 2 mm/min, 1 mm/min, and 0.5 mm/min, respectively, were utilized. The purpose of reducing the length of the open-hole specimens compared to the QI specimens was to maximize the number of specimens obtained from the manufactured plates. This approach aimed to save on raw materials and manufacturing costs due to limited resources. Similarly, the varying loading rates were chosen to ensure comparable strain rates among the different length specimens for both QI and Open-hole QI. It is worth noting that all the crosshead speeds used in the experiment fell within the quasi-static range. Consequently, the discrepancy in strain rate did not influence the progression of damage. Previous research [19] has already examined the impact of strain rate, revealing only minor alterations in failure mechanisms at high speeds (5–10 m/s). As a result, we do not anticipate any discernible effects on the failure mechanisms due to the slight difference in loading rates (2 mm/min vs. 1 mm/min). The strain was measured using an Imetrum video gauge system by tracing dotted patterns on the surface of the specimen. 

### 2.4. AE Device

A PAC data acquisition device (PCI-2 AE) was utilized to capture the AE signals. AE sensors were PAC R15 resonant-type, broadband, piezoelectric transducers, with an operating frequency range of 100–900 kHz. The gain selector and threshold value of the preamplifier were adjusted to 60 dB. The sampling rate for the test was set to 5 MHz. A pencil lead break process was utilized to calibrate measurements for each specimen. Following calibration, the AE signals were monitored through testing, as shown in Figure 4. Figure 5 provides a schematic definition of different AE signal features.

## 3. Results and Discussion

### 3.1. Mechanical Behavior and Resulted Failure Mechanisms

The results of the stress-strain analysis for both UD and QI configurations indicate a linear response at the beginning, a non-linear plateau, a second linear response, and the final load drop as shown in Figure 6 and Figure 7. The transparent characteristic of the glass layers allows for the monitoring of failure progression with the naked eye. The non-linear plateau is caused by the 0^o^ carbon plies’ fragmentation and stable pull-out of the fragmented carbon plies. After the fragmentation of the carbon plies reaches its maximum limit and they cannot carry the load, the remaining undamaged plies bear the load, leading to a rise in stress, as evidenced by the second linear response. The final load drop is because of the final catastrophic failure of the PDH. The QI open-hole specimens exhibit a linear response, which is succeeded by a final load drop, as indicated by the gross-section stress (refer to Figure 7). The CT-scan images (refer to Figure 8) for the investigated open-hole QI specimens taken at strain levels beneath the final load drop reveal that failure initiation occurs with fragmentations in the LSMs (carbon plies) at the notch’s edge. The resulting notch-insensitive behaviour was due to observed subcritical failure mechanisms, i.e., fragmentations in the LSMs and dispersed delaminations.

### 3.2. AE Results

Mechanical testing accompanied by in situ AE monitoring offers valuable insights into various aspects of failure evolution. Figure 9 shows a regular stress-time curve, as well as distributions of AE energy and cumulative AE energy. The initial major AE signals were observed nearby the non-linear plateau. In both UD and QI specimens, the AE events followed a similar trend, where the start of significant failure, namely, fragmentation and delamination, occurred close to the point where a significant change in stiffness occurs (knee point). However, for the open-hole specimen, the onset of failure was earlier than the final failure, caused by sub-critical failure resulting from stress concentration near the hole, as revealed in CT-scan results in Figure 8. The AE system was able to detect early failure even when there was no observable change in the stress-time graph. Previous studies [6,17] on glass/carbon hybrid laminates indicated that signals with high amplitude and energy are correlated with LSM fragmentation, whereas those with moderate values are typically related to delamination and interfacial debonding. However, as seen in the study, there is a significant variation in the AE event energy levels for the different specimens investigated, with average AE energy levels of 1100 aJ, 140 aJ, and 60 aJ for UD, QI, and open-hole QI specimens, respectively, indicating different failure mechanism intensities. In our previous study [5], it was shown that the fragmentations only occurred in the 0° carbon plies, as illustrated in Figure 10. The thickness of the adjacent 0° carbon plies in UD is twice the QI and open-hole QI configurations. Therefore, the resulting fragmentations are expected to be larger and generate a larger AE energy. In the next section, a multiparameter signal processing technique is applied to differentiate the failure mechanisms and their corresponding AE features and to highlight what are the best parameters for the classification of signals for different failure scenarios in thin-ply PDH laminates.

### 3.3. Clustering of the AE Signals and Their Correlation with the Failure Mechanisms

The objective of this section is to use multivariable clustering analysis of the AE signals to characterize the failure mechanisms. To achieve this, various AE features, such as count, amplitude, energy, duration, rise time, and three frequencies (initial, amplitude, and ringing) were employed. The AE features were initially subjected to principal component analysis (PCA) to decrease the data dimensions while retaining as much variability as possible [20,21]. Principal Component Analysis (PCA) is a powerful technique that enhances clustering algorithms by reducing the dimensionality of data [22]. It achieves this by transforming the original high-dimensional data into a lower-dimensional space, which brings several advantages. One of the primary objectives of PCA is to identify a set of new variables known as principal components that capture the maximum variance in the data. These components are derived through the calculation of linear combinations of the original features, ensuring that each component is orthogonal to the others. This orthogonal representation simplifies calculations, facilitates feature decorrelation, and reduces redundancy and noise in the data. The application of PCA leads to effective dimensionality reduction, improving both the computational complexity and the efficiency of clustering algorithms. By representing the data in a lower-dimensional space, the clustering algorithm can work with a reduced set of variables, resulting in faster calculations and enhanced scalability for large datasets. Additionally, PCA enables easier visualization of the data. Since the principal components retain the most significant variance, plotting the data points in this reduced-dimensional space allows for clearer visualization of clusters. This aids in understanding the inherent structure and patterns within the data, ultimately contributing to improved data analysis and better decision-making processes. As shown in Figure 11, the results of the PCA indicated that, in all specimens, the first two principal components accounted for over 70% of the total variability in the signal space. This suggests that a two-dimensional-projection of the first two PCA components retains over 70% of the variance, demonstrating the efficacy of PCA in reducing computational time and dimensions in the analysis.

Several clustering algorithms are commonly used to classify AE signals, including Self-Organizing Map (SOM), Hierarchical Model (HM), Gaussian Mixture Model (GMM), and Fuzzy C-Means (FCM) [21,23,24,25,26]. The number of clusters needs to be specified for these clustering algorithms. Gaussian mixture distribution algorithm and Davies Bouldin criteria were applied to figure out the optimal number of clusters, where the minimum Davies Bouldin index indicates the best solution. As shown in Figure 12, the Davies Bouldin index has the lowest value when the number of clusters is 2, indicating that this is the optimal number of clusters. This result is consistent with previously published studies [6,17] that also identified two clusters for induced failure in PDH composites. It is also consistent with the visual observations of the laminates investigated in the previous section, where the dominant failure mechanisms were found to be a combination of fragmentation and delamination.

Once the optimum clusters number was decided, the transformed data points from the PCA were grouped into two classes using three clustering methods (SOM, GMM, and HM). Previous research [17,23] has established that specific frequency ranges are associated with fibre breakage. While frequency is known to be a valuable AE feature for distinguishing between different failure mechanisms in composite materials, investigating the clustered data using all the three investigated clustering methods did not show this for the UD, QI, and open-hole specimens. As an example, Figure 13 shows that the AE signals associated with delamination and fragmentation overlap and do not exhibit a clear boundary. This was figured out by comparing each of the investigated AE features versus the clustered results by the combination of PCA and GMM. 

Analysis of the other AE features revealed that GMM produced the most favorable data distribution with minimal overlap between the clusters. Figure 14 depicts the distribution of three AE features (duration, energy, and amplitude) for the GMM-clustered signals, which were recognized as the most effective features by individual investigation of each AE feature versus the clustering results. Previous studies [17,18] have established that low energy and amplitude values are associated with delamination, while high energy and amplitude values are associated with fragmentation. The energy of AE signals provides valuable information about the severity and extent of carbon fibre fragmentation. Higher energy levels indicate a more severe fragmentation and a potential loss of load-bearing capacity. AE signal amplitude is another crucial parameter that can be utilized to analyze failure mechanisms in the PDH composites. The amplitude of AE signals correlates with the energy released during fibre fragmentation and the subsequent generation of associated stress waves. Higher amplitude levels indicate more substantial fibre fragmentation. The duration of AE signals provides insights into the time span of the failure mechanisms occurring in the PDH composites. In conventional composites, carbon fibre fracture typically produces short-duration AE signals due to the rapid fracture of individual fibres. However, the distribution range of AE signal durations for fragmentation is higher than the dispersed delamination, indicating the presence of shorter-duration signals that are associated with stable crack propagation. 

Table 2 also shows the average values for three AE features presented in Figure 14 for fibre fragmentation. There are variations in both the intensity and distribution of the failure modes among different types of specimens. For the UD, fragmentations produced the highest energy level, which can be related to the thicker carbon block consisting of two layers of TR30 compared to the QI with only one layer of TR30. The fragmentations in the open hole QI exhibited the lowest energy level, likely because of the local fragmentations near the hole that do not generate a high energy level. Despite some areas of overlap among the AE features shown in Figure 14, the normal distribution of these parameters is demonstrated in Figure 15. The distribution graphs indicate a small area of overlap for the three AE features. Consequently, duration, amplitude, and energy are crucial AE features for differentiating between delamination and fibre fragmentation in all three specimens. Figure 15 shows different levels of overlap for the ply fragmentations and delamination among the investigated specimens, reflecting the diversity of the intensity and size of these failure modes. 

## 4. Conclusions

This study aimed to assess the potential of AE-based characterization for monitoring failure mechanisms in a comprehensive range of thin-ply glass/carbon PDH composites. The key findings and conclusions drawn from this study are as follows:CT-scan and visual images confirmed that the fragmentations and dispersed delamination were the dominant failure modes observed in the tested specimens. These failure mechanisms resulted in a successful pseudo-ductile behaviour for the UD and QI laminates, as well as notch-insensitivity in the open-hole QI laminate.The AE signals generated from the UD laminate have a higher energy level than the QI and open-hole laminates, which is due to the thicker block of adjacent 0° carbon plies, causing larger fragmentations.Applying GMM to cluster the AE signals uncovered two distinct types of events: those with high energy, amplitude, and duration, which were linked to carbon ply fragmentation, and those with low values of those parameters, which were connected to delamination of the glass/carbon interface. It was also found that, in contrast to the literature, high-frequency AE signals are not necessarily linked with carbon fibre fragmentation, and they are also linked with delamination, with no distinguishable ranges.The proposed clustering method and AE-based approach were found to be a simple yet robust tool for characterizing failure mechanisms in hybrid laminates and can serve as an early warning and failure detection method. However, it should be noted that the AE characteristics may vary depending on the stacking sequence and configurations, making it challenging to generalize the clustering criteria for different PDH configurations.

## Figures and Tables

**Figure 1 sensors-23-05244-f001:**
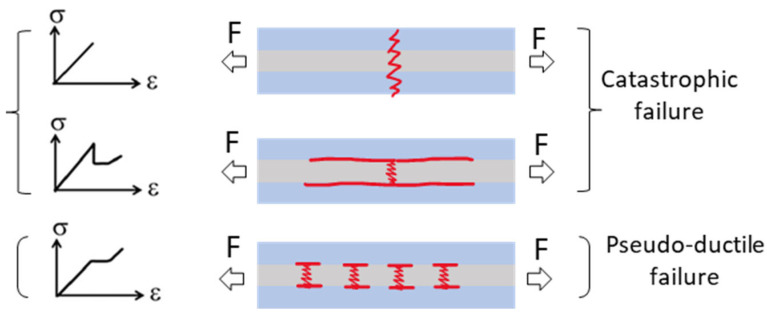
Three potential modes of failures in a hybrid laminate.

**Figure 2 sensors-23-05244-f002:**
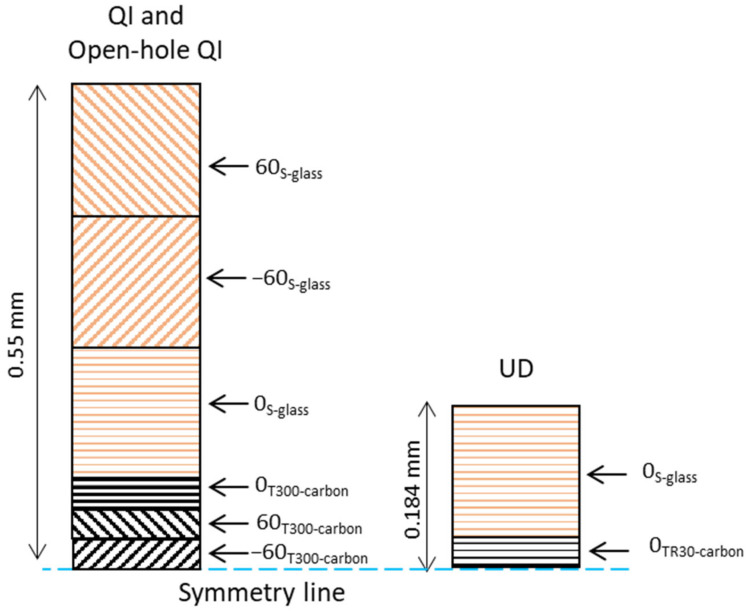
Schematics of the lay-up arrangements: the plies made of S-glass are represented by the color brown, while plies made of T300-carbon/TR30-carbon are represented by black.

**Figure 3 sensors-23-05244-f003:**
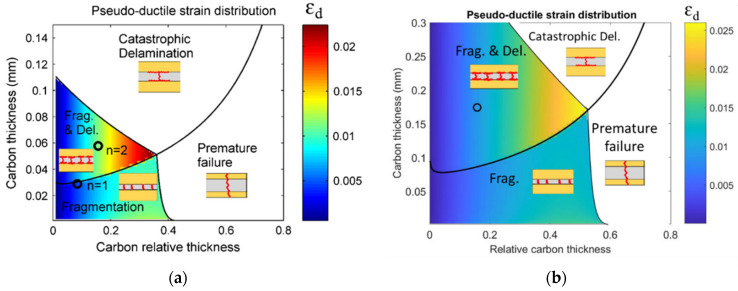
Failure mode maps in the investigated configurations, UD (**a**) and QI (**b**).

**Figure 4 sensors-23-05244-f004:**
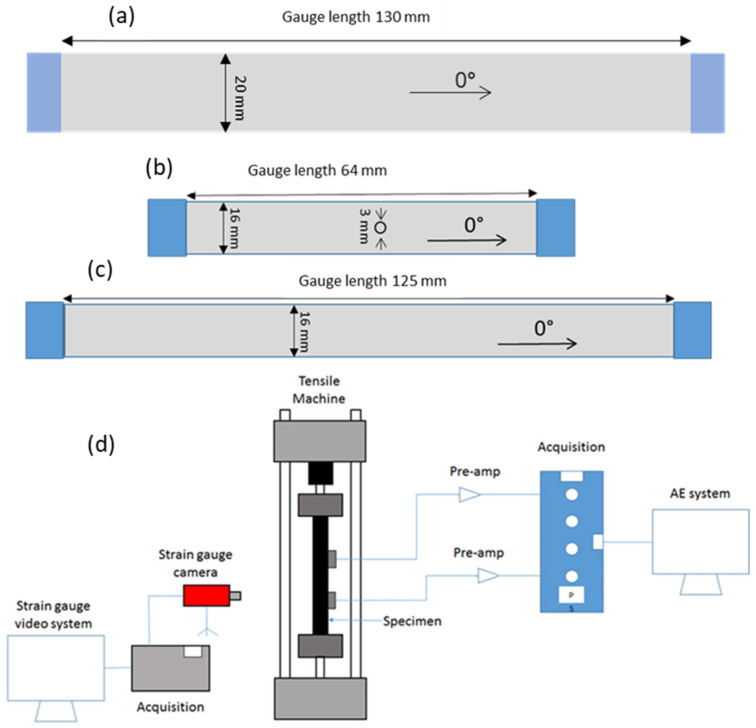
Schematics of (**a**) the UD specimen, (**b**) open-hole QI specimen, (**c**) QI specimen, as well as (**d**) the experimental setup schematic.

**Figure 5 sensors-23-05244-f005:**
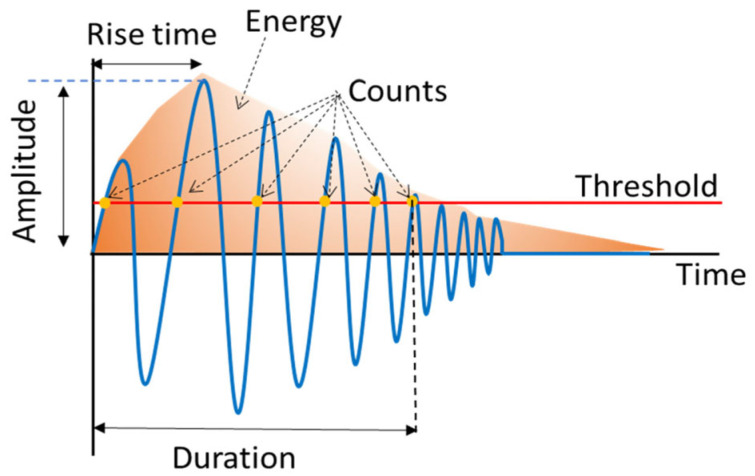
Definitions of parameters used in acoustic emission analysis.

**Figure 6 sensors-23-05244-f006:**
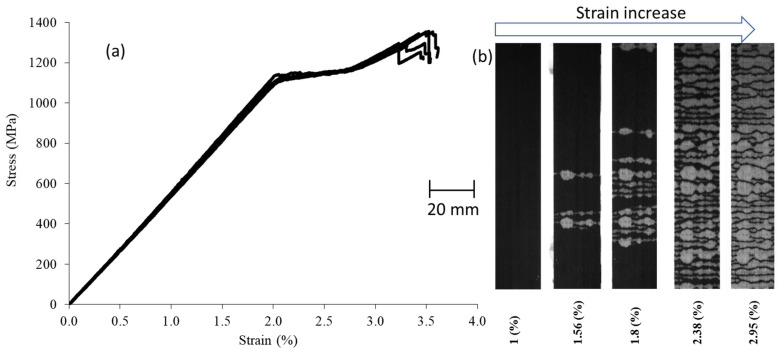
(**a**) Tensile test results for a UD specimen. (**b**) Its top view in black and white colour.

**Figure 7 sensors-23-05244-f007:**
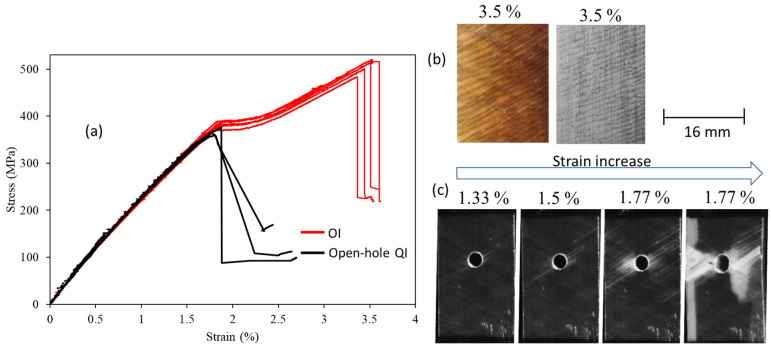
(**a**) Tensile test results for QI and open-hole QI specimens. Top views of (**b**) the surface and 0° carbon layer for the QI specimen. (**c**) The surface of the open-hole QI specimens in black and white color.

**Figure 8 sensors-23-05244-f008:**
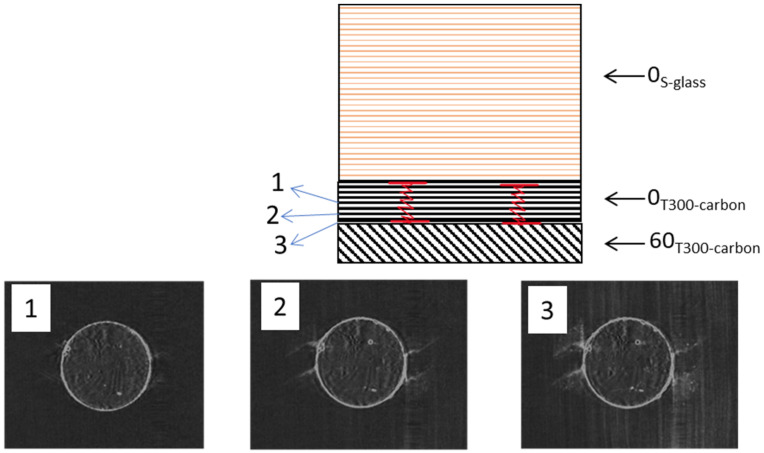
The CT-scan results of the QI open-hole laminate at strain level of 1.62% were captured from three different locations (1, 2, and 3), viewed from the top.

**Figure 9 sensors-23-05244-f009:**
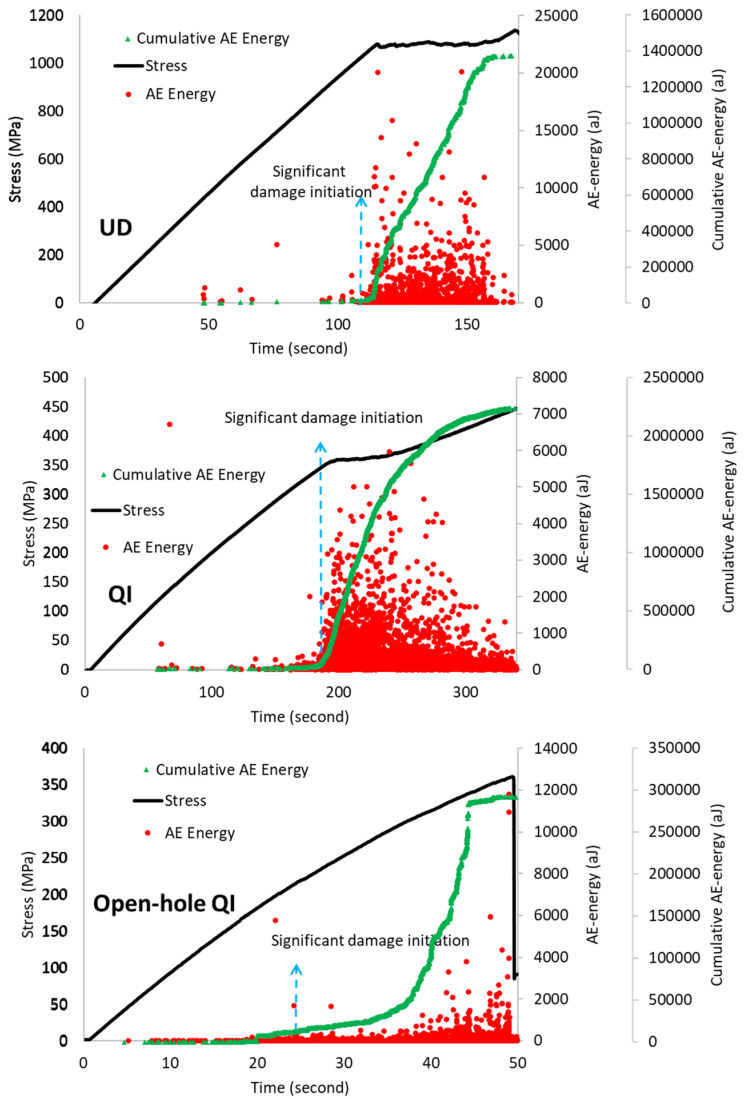
Stress–time and AE energy and cumulative AE energy distributions for the UD, QI, and open-hole QI specimens.

**Figure 10 sensors-23-05244-f010:**
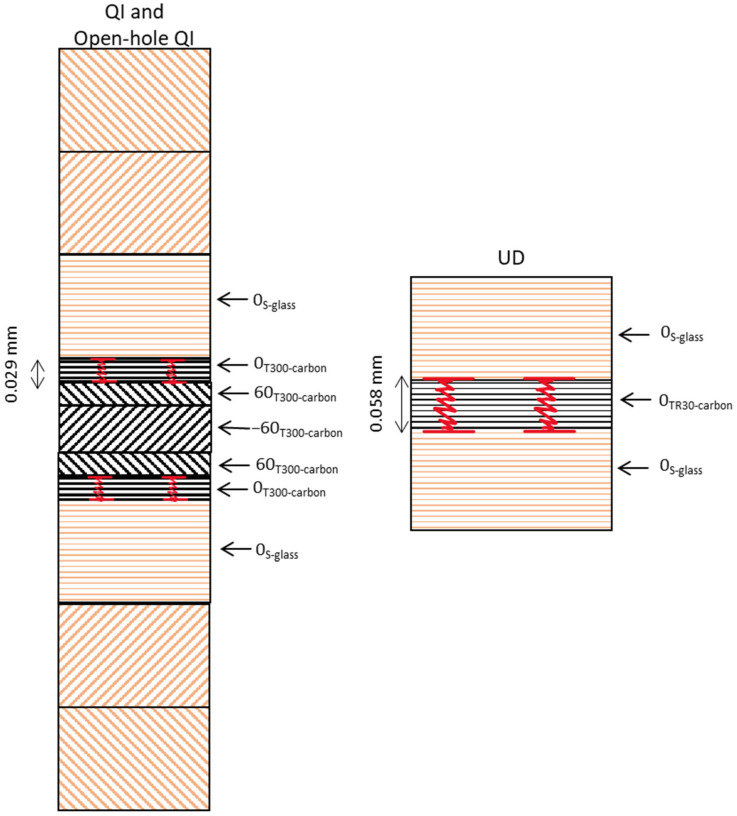
Schematic of the failure mechanisms for the UD and QI configurations. Fragmentations and dispersed delamination are shown by the red lines.

**Figure 11 sensors-23-05244-f011:**
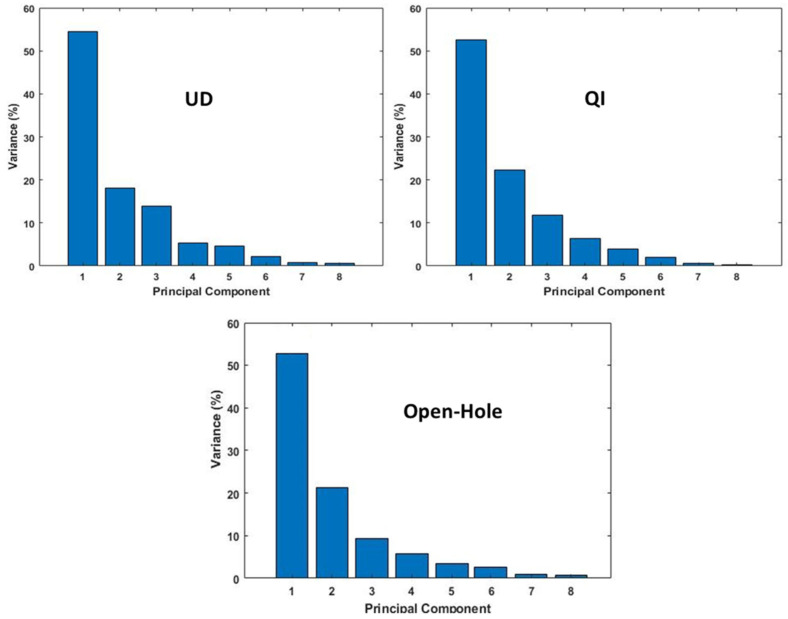
The variability percentage of each principal component for each of the specimens.

**Figure 12 sensors-23-05244-f012:**
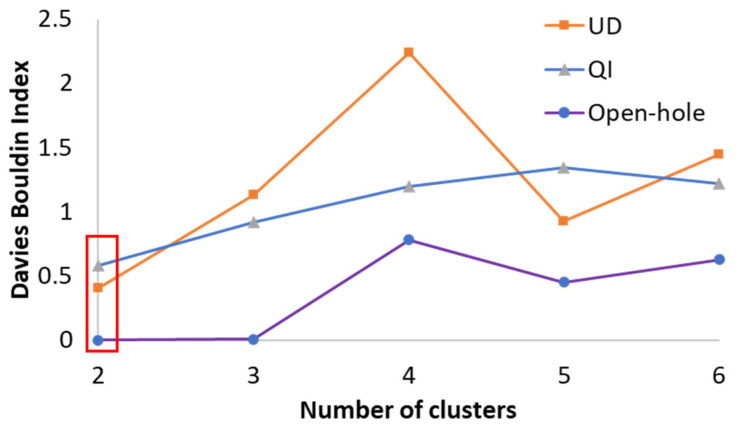
Davies Bouldin index plot, the red box shows the minimum index for the specimens.

**Figure 13 sensors-23-05244-f013:**
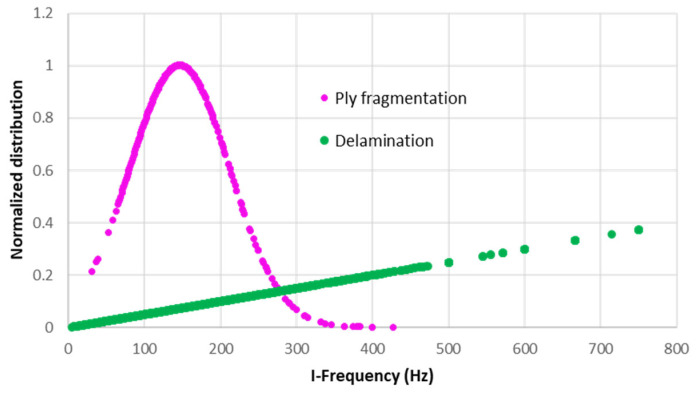
Distribution of the initial frequency for the QI specimen.

**Figure 14 sensors-23-05244-f014:**
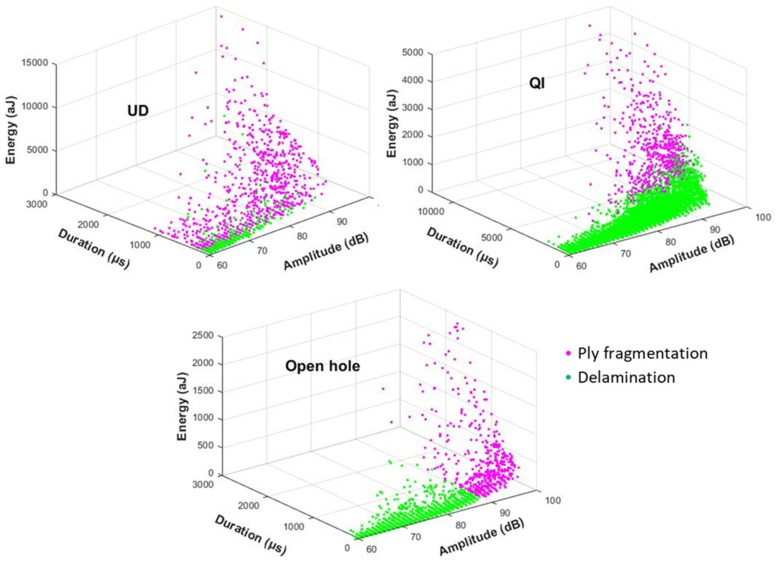
GMM clustering of AE duration, amplitude, and energy.

**Figure 15 sensors-23-05244-f015:**
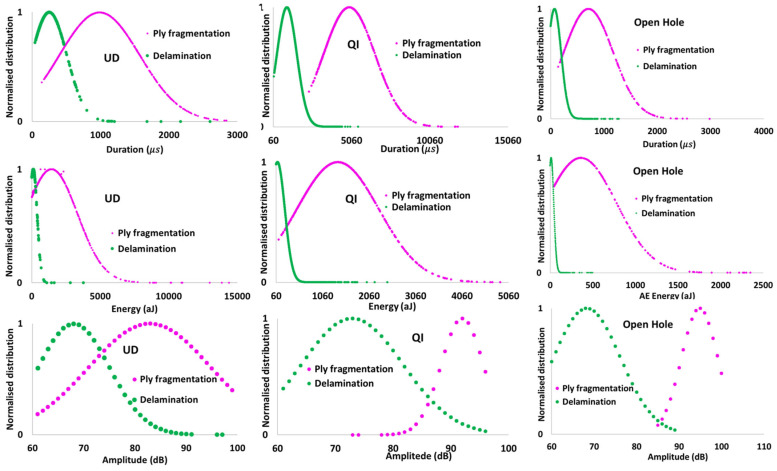
Normalised distribution of the AE signals: duration, energy, and amplitude domain.

**Table 1 sensors-23-05244-t001:** Characteristics of the utilized prepregs.

Prepreg Type	TR30/epoxy [6]	S-glass/913epoxy [6]	T300/epoxy [17]
Fibre modulus (GPa)	234	88	230
Prepreg fibre direction Modulus, E11 (GPa)	101.7	45.7	101.7
Fibre failure strain (%)	1.9	5.5	1.5
Cured nominal thickness (mm)	0.029	0.155	0.029
Fibre mass per unit area (g/m^2^)	21.2	190	22
Fibre volume fraction (%)	41	50	43
Supplier	Sky Flex	Hexcel	Sky Flex

**Table 2 sensors-23-05244-t002:** GMM clustering of the AE signals for fibre fragmentation.

	Number of Fragmentation	Average Energy (aJ)	Average Duration (μs)	Average Amplitude (dB)
UD	684	1902	988	83
QI	765	1266	4921	92
Open Hole	511	465	710	95

## Data Availability

Not applicable.

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
