# Peer review of "Multivariable Signal Processing for Characterization of Failure Modes in Thin-Ply Hybrid Laminates Using Acoustic Emission Sensors"

_sensors, 2023, doi:10.3390/s23115244_

Round 1

Reviewer 1 Report

1."For the UD, QI, and open-hole QI laminates, crosshead speeds of 2 mm/min, 1 mm/min, and 0.5 mm/min, respectively, were utilized." Different speeds may produce different results; why use different speeds? Please explain.

2. The gauge length varies for each configuration. Will this have an impact on the results?

3. In Figure 9, stress–time, AE energy, and cumulative AE energy distributions for each configuration should be placed in one image.

4. In Figure 13, there needs a legend.

5. The analysis could be more explicit in Figure 13, Table 3, and Figure 14. Authors should analyze the AE signals and their correlation with the failure mechanisms in detail.

6. Some conclusions lack support, such as "It was also found that in contrast to the literature, high-frequency AE signals are not linked necessarily linked with carbon fibre fragmentation, and they are also linked with delamination, with no distinguishable ranges."

7. "The proposed clustering method and AE-based approach were found to be a simple yet robust tool." However, what is the difference between this method and the literature [6,17]? Or is it simply using the method of the literature?

Author Response

The response file is attached.

1."For the UD, QI, and open-hole QI laminates, crosshead speeds of 2 mm/min, 1 mm/min, and 0.5 mm/min, respectively, were utilized." Different speeds may produce different results; why use different speeds? Please explain.

  1. The gauge length varies for each configuration. Will this have an impact on the results?

Comments 1 and 2 are related. The explanation is included in the revised version, page 5, test procedure section.

 “The purpose of reducing the length of the open-hole samples compared to the QI samples was to maximize the number of samples obtained from the manufactured plates. This approach aimed to save on raw materials and manufacturing costs due to limited resources. Similarly, the varying loading rates were chosen to ensure comparable strain rates among the different length specimens for both QI and Open-hole QI. It is worth noting that all the crosshead speeds used in the experiment fell within the quasi-static range. Consequently, the discrepancy in strain rate did not influence the progression of damage. Previous research [19] has already examined the impact of strain rate, revealing only mi-nor alterations in failure mechanisms at high speeds (5-10 m/s). As a result, we do not anticipate any discernible effects on the failure mechanisms due to the slight difference in loading rates (2 mm/min vs. 1 mm/min).”

[19] Fotouhi, M., Fuller, J., Longana, M., Jalalvand, M. and Wisnom, M.R., 2019. The high strain rate tension behaviour of pseudo-ductile high performance thin ply composites. Composite Structures, 215, pp.365-376.

  1. In Figure 9, stress–time, AE energy, and cumulative AE energy distributions for each configuration should be placed in one image.

It is modified.

  1. In Figure 13, there needs a legend.

It is modified.

  1. The analysis could be more explicit in Figure 13, Table 3, and Figure 14. Authors should analyze the AE signals and their correlation with the failure mechanisms in detail.

More information is provided in this section on connecting the AE results and associated failure mechanisms.

  1. Some conclusions lack support, such as "It was also found that in contrast to the literature, high-frequency AE signals are not linked necessarily linked with carbon fibre fragmentation, and they are also linked with delamination, with no distinguishable ranges."

This was provided in Figure 13 (previously Figure 12) showing the overlap in the Frequency of delamination and fragmentation.  However, it is elaborated more in the revised version to make it clear.

  1. "The proposed clustering method and AE-based approach were found to be a simple yet robust tool." However, what is the difference between this method and the literature [6,17]? Or is it simply using the method of the literature?

The literature [6, 17] demonstrated the effectiveness of the AE technique using single-parameter amplitude and energy distributions. However, in this paper, multivariable analysis tools, PCA and SOM, are employed to categorize the AE signals and assess the failure mechanisms and their progression in a comprehensive range of thin-ply PDH composite materials to identify the main AE features associated with different ranges of these failure mechanisms varying in size and intensity.

Reviewer 2 Report

1.Line 56:tiple fractures occur in the LSM, leading to dispersed delamination [14,15,16]. The format of citation is not correct.

2. In the illustration of the Figure.9, the identification of energy and cumulative energy must be given? What is the difference of these two energy,Why utilize both of them,these points must be clarified.

3. Fig or Figure? Must be utilized in the whole manuscript, For exmaple it has been used in “Fig.1” in Line 55, and used in “Figure 2” in line 107.

The abstract should be written in the past tense.

Author Response

1.Line 56:tiple fractures occur in the LSM, leading to dispersed delamination [14,15,16]. The format of citation is not correct.

It is corrected.

  1. In the illustration of the Figure.9, the identification of energy and cumulative energy must be given? What is the difference of these two energy,Why utilize both of them,these points must be clarified.

One distribution is AE energy and the other is cumulative AE energy. The reason for using both is to show the AE energy from each failure and also the cumulative energy from the accumulated damage. The picture is modified and both distributions are shown in 1 graph now.

  1. Fig or Figure? Must be utilized in the whole manuscript, For exmaple it has been used in “Fig.1” in Line 55, and used in “Figure 2” in line 107.

It is modified.

Comments on the Quality of English Language

The abstract should be written in the past tense.

This is modified. The English is reviewed by a native speaker to avoid any errors.

Reviewer 3 Report

This is an interesting study about damage characterization of composite laminates with acoustic emission sensors. The text is concise and clear with good quality figures. The authors are presenting good correlation between AE features and the two main failure modes, ply fragmentation and delamination.

Section 3.3 requires more details. In particular,

- It is not clear in the study how the authors selected the energy, duration and amplitude for the multivariable analysis, while excluding other features, such as rise time, RA value and frequency. Can you provide further justification? Some analysis results will be helpful

- How much the PCA has improved the results? Can you provide evidence or further justification? It is important for the reader to understand to what extent this needs to be a standard process before clustering.

Page 14, line 287 “… linked necessarily linked…” The word ‘linked’ is repeated

Author Response

This is an interesting study about damage characterization of composite laminates with acoustic emission sensors. The text is concise and clear with good quality figures. The authors are presenting good correlation between AE features and the two main failure modes, ply fragmentation and delamination.

Section 3.3 requires more details. In particular,

- It is not clear in the study how the authors selected the energy, duration and amplitude for the multivariable analysis, while excluding other features, such as rise time, RA value and frequency. Can you provide further justification? Some analysis results will be helpful

We looked at different features, and energy, duration and amplitude provide the best clustering results. Further information and figure 11 in this regard.

- How much the PCA has improved the results? Can you provide evidence or further justification? It is important for the reader to understand to what extent this needs to be a standard process before clustering.

Further information is provided in the relevant section.

PCA has an important effect on clustering by reducing data dimension while keeping essential information to find patterns in the high-dimensional data set. Otherwise, recognising the most significant features for classification can be challenging. In other words, PCA and clustering techniques (GMM in this work) are not directly comparable, but they can be complementary and used in combination in certain scenarios. Besides, clustering techniques can be affected by variations in feature scales. PCA transforms the data into a new coordinate system, and this enhances clustering by equalizing feature scales.

Page 14, line 287 “… linked necessarily linked…” The word ‘linked’ is repeated

It is modified.

Round 2

Reviewer 1 Report

No Comments.

Reviewer 3 Report

Authors have addressed my comments. I recommend publication.